# TSCCA: A tensor sparse CCA method for detecting microRNA-gene patterns from multiple cancers

Wenwen Min[1,2,3], Tsung-Hui Chang[1,2], Shihua Zhang[4,5,6,7]*, Xiang Wan[1]*

**1** Shenzhen Research Institute of Big Data, Shenzhen, China, **2** School of Science and Engineering, The Chinese University of Hong Kong, Shenzhen, China, **3** University of Science and Technology of China, Hefei, China, **4** NCMIS, CEMS, RCSDS, Academy of Mathematics and Systems Science, Chinese Academy of Sciences, Beijing, China, **5** School of Mathematical Sciences, University of Chinese Academy of Sciences, Beijing, China, **6** Center for Excellence in Animal Evolution and Genetics, Chinese Academy of Sciences, Kunming, China, **7** Key Laboratory of Systems Biology, Hangzhou Institute for Advanced Study, University of Chinese Academy of Sciences, Chinese Academy of Sciences, Hangzhou, China

* zsh@amss.ac.cn (SZ); wanxiang@sribd.cn (XW)

**Data Availability Statement:** We used the biological data from 33 TCGA cancer types available from the Broad GDAC Firehose website (http://firebrowse.org/, accessed 28 January 2016). The code of TSCCA is available from https://github.com/wenwenmin/TSCCA.

## Abstract

Existing studies have demonstrated that dysregulation of microRNAs (miRNAs or miRs) is involved in the initiation and progression of cancer. Many efforts have been devoted to identify microRNAs as potential biomarkers for cancer diagnosis, prognosis and therapeutic targets. With the rapid development of miRNA sequencing technology, a vast amount of miRNA expression data for multiple cancers has been collected. These invaluable data repositories provide new paradigms to explore the relationship between miRNAs and cancer. Thus, there is an urgent need to explore the complex cancer-related miRNA-gene patterns by integrating multi-omics data in a pan-cancer paradigm. In this study, we present a tensor sparse canonical correlation analysis (TSCCA) method for identifying cancer-related miRNA-gene modules across multiple cancers. TSCCA is able to overcome the drawbacks of existing solutions and capture both the cancer-shared and specific miRNA-gene co-expressed modules with better biological interpretations. We comprehensively evaluate the performance of TSCCA using a set of simulated data and matched miRNA/gene expression data across 33 cancer types from the TCGA database. We uncover several dysfunctional miRNA-gene modules with important biological functions and statistical significance. These modules can advance our understanding of miRNA regulatory mechanisms of cancer and provide insights into miRNA-based treatments for cancer.

## Author summary

MicroRNAs (miRNAs) are a class of small non-coding RNAs. Previous studies have revealed that miRNA-gene regulatory modules play key roles in the occurrence and development of cancer. However, little has been done to discover miRNA-gene regulatory modules from a pan-cancer view. Thus, it is urgently needed to develop new methods to explore the complex cancer-related miRNA-gene patterns by integrating multi-omics

**Funding:** The work of X.W. was supported by the Key-Area Research and Development Program of Guangdong Province of China [2020B0101350001]. The work of W.M. was supported by the National Science Foundation of China [61802157], the Natural Science Foundation of Jiangxi Province of China [20192BAB217004] and the China Postdoctoral Science Foundation [2020M671902]. The work of T-H.C. was supported by the Open Research Fund from Shenzhen Research Institute of Big Data [2019ORF01002] and the National Science Foundation of China [61731018]. The work of S.Z. was supported by the National Science Foundation of China [11661141019, 61621003], the National Ten Thousand Talent Program for Young Top-notch Talents, the National Key Research and Development Program of China [2019YFA0709501] and the CAS Frontier Science Research Key Project for Top Young Scientist [QYZDB-SSW-SYS008]. The funders had no role in study design, data collection and analysis, decision to publish, or preparation of the manuscript.

**Competing interests:** The authors have declared that no competing interests exist.

data of multi-cancers. To build the connections between miRNA-gene regulatory modules across different cancer types, we propose a tensor sparse canonical correlation analysis (TSCCA) method. Our specific contributions are two-fold: (1) We propose a sparse statistical learning model TSCCA and an efficient block-coordinate descent algorithm to solve it. (2) We apply TSCCA to a multi-omics data set of 33 cancer types from TCGA and identify some cancer-related miRNA-gene modules with important biological functions and statistical significance.

## Introduction

Cancer is a complex and heterogeneous disease and the second leading cause of death worldwide [1, 2]. Although medical advances have made possible earlier diagnosis and more effective treatments, researchers still face many critical challenges for cancer drug resistance, combinatorial drug treatment optimization and personalized cancer therapy design and so on [3, 4]. A number of studies have been conducted to understand the mechanisms underlying the cancer development for better prevention and treatment.

In the past decade, an increasing number of studies have reported that abnormal microRNAs (miRNAs) play important roles in the occurrence and development of cancer [5, 6], and some miRNAs can be used as drug targets for cancer treatment [7, 8]. miRNA is a type of small non-coding RNAs with about 20 bases, which regulates gene expression during post-transcriptional processes [9]. In cancer cells, miRNAs have been found to be heavily dysregulated [8]. Thus, they are potential candidates for prognostic biomarkers and therapeutic targets in cancer. For example, Yang *et al.* have reported that miR-506 plays essential roles in the pathogenesis of ovarian cancer, which can be considered as a potential therapeutic interest [7]. Moreover, Lai *et al.* outlined some miRNAs as monotherapy or adjuvant therapy from a systems biology perspective [8].

Since miRNAs were found, researchers have studied the regulatory mechanisms between miRNAs and genes comprehensively. For example, sequence-based methods have been proposed to predict their regulatory relationships [10, 11]. However, such methods fail to capture the context-specific miRNA-gene regulatory relationship. With the development of miRNA sequencing technology, a huge number of miRNA expression data of multi-species have been accumulated (e.g., those in the Gene Expression Omnibus database repository [12]). The Cancer Genome Atlas (TCGA) [13] and NCI-60 [14] allow us to obtain matched miRNA and mRNA expression data in certain cancers. These invaluable database repositories provide new paradigms to explore context-specific miRNA-gene regulatory relationship. Several computational methods have been proposed on the basis of modular structure identification [15–21]. Zhang *et al.* developed a joint non-negative matrix factorization method to discover miRNA-gene co-modules in ovarian cancer [15]. However, the strength of miRNA-gene relationship in the identified modules by it is still unclear and the algorithm therein has a high computational complexity. Min *et al.* developed a simple two-step method for the same task [16]. This method firstly reconstructs a sparse miRNA-gene regulation matrix by integrating miRNA and mRNA expression data and prior miRNA group information. Then, a bi-clustering method based on a sparse matrix factorization is used to cluster the regulation matrix for discovering miRNA-gene modules. Yoon *et al.* (2019) also developed a bi-clustering method to identify condition-specific modules by integrating the gene expression and miRNA sequence-specific targets information [21]. Although these methods can discover miRNA-gene modules

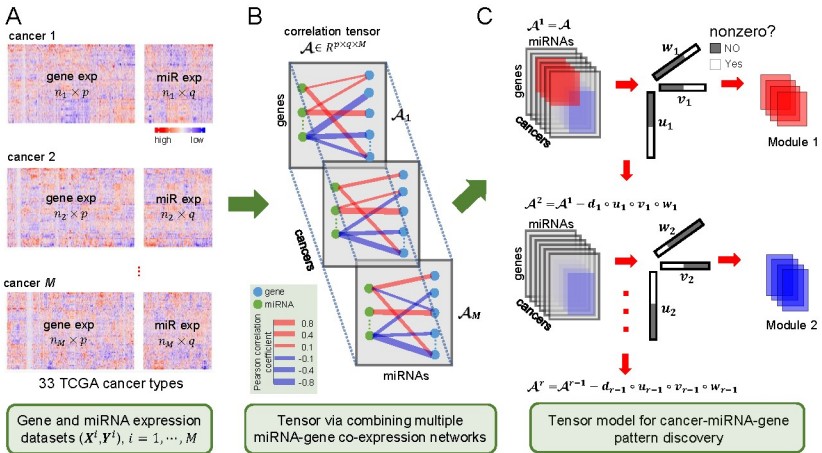

**Fig 1. Illustration of TSCCA to identify cancer-related miRNA-gene functional modules.** (A) Prepare the matched miRNA and gene expression data of 33 cancer types from TCGA. (B) Compute a cancer-miRNA-gene Pearson correlation tensor $\mathcal{A} \in \mathbb{R}^{p \times q \times M}$, where $p$, $q$ and $M$ represent the number of genes, miRNAs and cancers respectively. (C) Estimate multiple sparse latent factors ($u_i$, $v_i$ and $w_i$, $i = 1, \cdots, r$) and these non-zero genes in $u_i$, non-zero miRNAs in $v_i$ and non-zero cancers in $w_i$ are considered as a cancer-miRNA-gene module.

for one cancer or tissue to some extent, they fail to identify cancer-specific and shared miRNA-gene modules when integrating multiple cancer data.

Recently, some studies have focused on the integrative analysis of multiple omics data from multiple cancers [22–26]. For example, Tan *et al*. systematically investigated the positive correlation between miRNAs and genes in multiple human cancers [26]. However, little has been done to discover miRNA-gene regulatory modules from a pan-cancer view. Therefore, it is urgently needed to develop new methods to explore the complex cancer-related miRNA-gene patterns by integrating multi-omics data of multi-cancers.

In this study, we present a tensor sparse canonical correlation analysis (TSCCA) method for the explorative analysis of matched miRNA and gene expression data of multiple cancers with a focus on identifying cancer-specific and shared miRNA-gene co-expressed modules (Fig 1). TSCCA first calculates a cancer-miRNA-gene correlation tensor which is a "3D" array with gene, miRNA and cancer dimensions (Fig 1B). Then it decomposes the correlation tensor into a number of latent factors ($u_i$, $v_i$ and $w_i$, $i = 1, \cdots, r$) that represent major patterns of variation in the tensor data (Fig 1C). The scores of $u_i$, $v_i$ and $w_i$ indicate the relative contribution of genes, miRNAs and cancers, respectively. Based on their non-zero elements of $u_i$, $v_i$ and $w_i$ for any $i$, we can discover a cancer-miRNA-gene module. In short, our main contributions are two-fold: (1) We design a statistical learning model TSCCA, which is equivalent to a $\ell_0$-norm constrained tensor-based model, and develop an efficient block-coordinate descent algorithm to solve it. (2) We apply TSCCA to a multi-omics data set of 33 cancer types from TCGA database and discover some dysfunctional miRNA-gene modules with important biological functions and statistical significance.

## Materials and methods

### Biological data

**TCGA data.** We used the biological data from 33 TCGA cancer types available from the Broad GDAC Firehose website (http://firebrowse.org/, accessed 28 January 2016). For each cancer type, we downloaded the processed (Level 3) mRNA-seq and miRNA-seq data, and

clinical data. Before applying our method, we implemented multi-step data preprocessing for each cancer data set: (1) We removed those genes and miRNAs, which are expressed in less than 5% samples; (2) Missing elements were imputed using the $k$-nearest neighbor method by using the R package "impute". (3) The expression values were log2 transformed and scaled with zero mean and unit standard deviation for every gene/miRNA. (4) Differential gene expression analysis was carried out by using the Wilcox test for each gene when the cancer type contains more than 5 normal samples. We found 7889 pan-cancer significant differentially expressed genes in more than 15 cancers with Benjamini-Hochberg (BH) adjusted $P < 0.05$ and the detailed results are shown in S1 Table. Finally, we obtained the matched mRNA and miRNA expression data of 33 cancer types including 9645 cancer samples, 7889 genes and 523 miRNAs (Fig 2A and S2 Table). To further analyze the biological functions of the cancer-miRNA-gene module, we also downloaded the following data sets:

**miRNA family database**. We downloaded a miRNA family data set from miRbase database [9]. A miRNA family contains a set of miRNAs.

**miRNA-gene interaction network data**. We collected an experimentally validated miRNA-gene interaction network data set from miRTarBase database [27].

**Gene interaction network data**. We downloaded a protein-protein interaction (PPI) network data set from the Pathway-Commons database [28]. A gene interaction network was constructed by the PPI network.

**Cancer gene and miRNA sets**. We collected a cancer gene set data from the allOnco database (http://www.bushmanlab.org/links/genelists) and a cancer miRNA set data from http://mircancer.ecu.edu/ [29].

**Gene functional annotations**. We also downloaded multiple gene functional annotations including GO biological processes (GOBP), KEGG and reactome pathways from Molecular Signatures Database (MSigDB) [30].

## Sparse CCA

Canonical Correlation Analysis (CCA) is a common statistical learning method for analyzing pairwise data. It learns a projection for both representations such that they are maximally correlated in the dimensionality-reduced space. Suppose $X \in \mathbb{R}^{n \times p}$ with $n$ samples and $p$ features and $Y \in \mathbb{R}^{n \times q}$ with $n$ samples and $q$ features represent two omics data from a single cancer and their columns of $X$ and $Y$ are centered and scaled with zero mean and unit variance. Then, the CCA model can be written as follows:

$$\begin{aligned} &\underset{u,v}{\text{maximize}} \quad u^T X^T Y v \\ &\text{subject to} \quad u^T X^T X u = 1, v^T Y^T Y v = 1. \end{aligned} \tag{1}$$

Suppose $X^T X = I$ and $Y^T Y = I$, where $I$ is the identity matrix. Then the above model reduces to:

$$\begin{aligned} &\underset{u,v}{\text{maximize}} \quad u^T X^T Y v \\ &\text{subject to} \quad u^T u = 1, v^T v = 1. \end{aligned} \tag{2}$$

which was called as the diagonal CCA whose performance is usually better than the traditional CCA in high-dimensional data [31, 32]. However, the classical CCA leads to non-sparse canonical vectors. It is difficult to select features and interpret in biology. To this end, a large

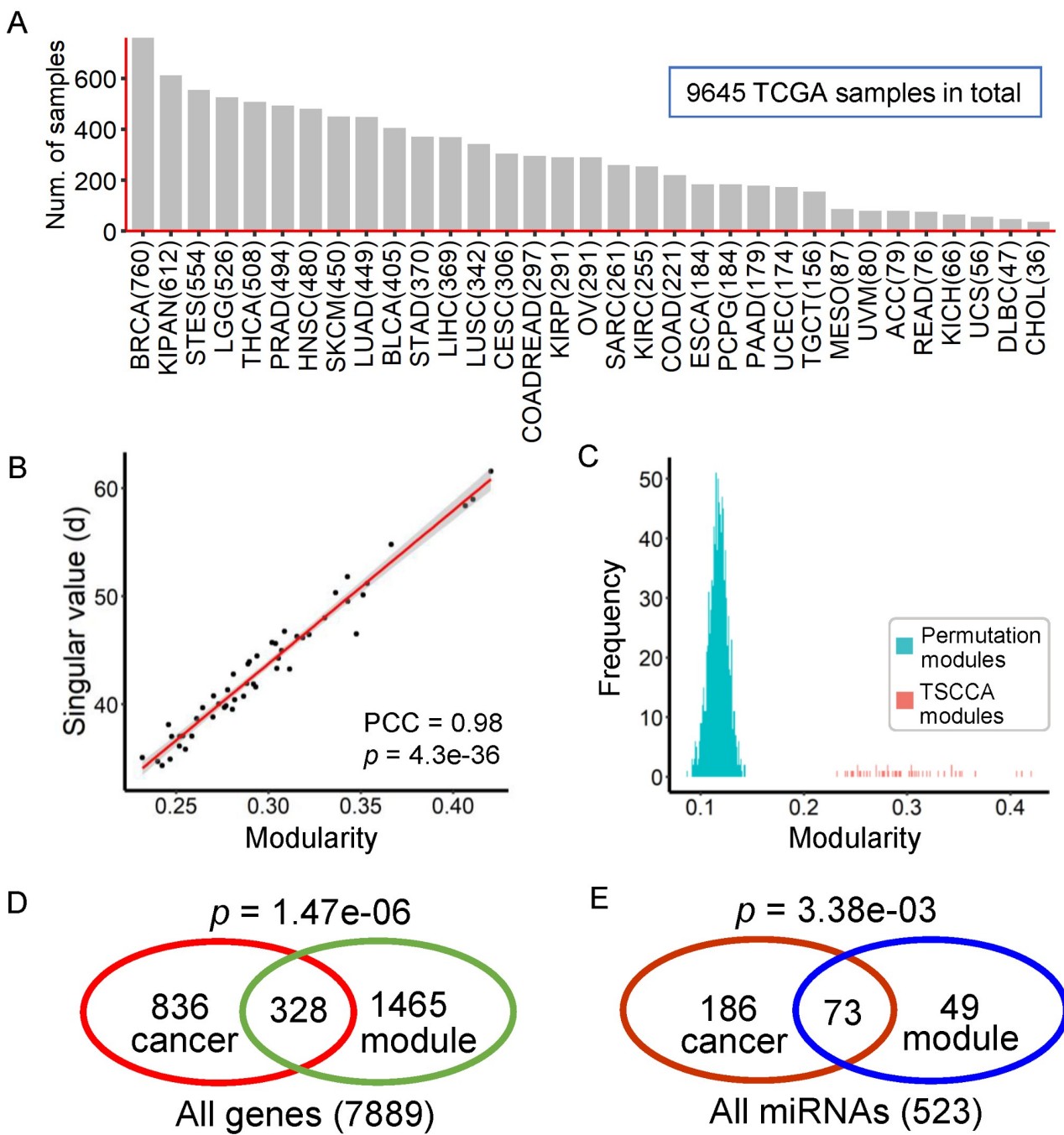

**Fig 2. Application to the TCGA data from multiple cancers.** (A) Number of cancer patients or samples on 33 cancer types from TCGA in this study. (B) Correlation between the modularity scores of identified modules (*y*-axis) and the corresponding singular values (objective function values) (*x*-axis) with PCC *r* = 0.98. (C) Distribution of modularity scores. The modularity scores of identified modules are significantly greater than those of random ones (Permutation test *P* < 0.05/50 for each identified module). (D) Among the 1793 genes from all the identified modules, 328 are reported to be related with cancer (Hypergeometric test *P* = 1.47e-06). (E) Among the 122 miRNAs from all the identified modules, 73 are reported to be related with cancer (Hypergeometric test *P* = 3.38e-03).

number of sparse CCA models have been proposed to obtain sparse canonical vectors by using different penalty functions [16, 33–37]. Specifically, a sparse CCA (SCCA) with $\ell_0$-norm constraint [35] can be formulated into the following optimization problem:

$$
\begin{aligned}
\underset{\boldsymbol{u},\boldsymbol{v}}{\text{maximize}} \quad & \boldsymbol{u}^T \boldsymbol{X}^T \boldsymbol{Y} \boldsymbol{v} \\
\text{subject to} \quad & \|\boldsymbol{u}\|_0 \le k_u, \|\boldsymbol{v}\|_0 \le k_v, \\
& \boldsymbol{u}^T \boldsymbol{u} = 1, \boldsymbol{v}^T \boldsymbol{v} = 1,
\end{aligned}
\tag{3}
$$

where $k_u$ and $k_v$ are two parameters to control the sparsity of canonical vectors ($\boldsymbol{u}$ and $\boldsymbol{v}$), and $\|\boldsymbol{u}\|_0$ is the number of non-zero elements in the $\boldsymbol{u}$.

## Proposed tensor sparse CCA (TSCCA)

Let $\boldsymbol{X}^i \in \mathbb{R}^{n_i \times p}$ with $n_i$ samples and $p$ genes and $\boldsymbol{Y}^i \in \mathbb{R}^{n_i \times q}$ with $n_i$ samples and $q$ miRNAs be the matched gene and miRNA expression matrices of cancer $i$ ($i = 1, \cdots, M$). Each column of them is normalized with zero-mean and unit-variance (Fig 1A). To capture invariant miRNA-gene co-expressed pattern for different cancers, we propose a tensor-based method to integrate miRNA and gene expression data from multiple cancers by weighting each cancer as follows:

$$
\begin{aligned}
\underset{\boldsymbol{u},\boldsymbol{v},\boldsymbol{w}}{\text{maximize}} \quad & \sum_{i=1}^{M} w_i (\boldsymbol{u}^T \boldsymbol{X}^{i^T} \boldsymbol{Y}^i \boldsymbol{v}) \\
\text{subject to} \quad & \|\boldsymbol{u}\|_0 \le k_u, \|\boldsymbol{v}\|_0 \le k_v, \|\boldsymbol{w}\|_0 \le k_w, \\
& \boldsymbol{u}^T \boldsymbol{u} = 1, \boldsymbol{v}^T \boldsymbol{v} = 1, \boldsymbol{w}^T \boldsymbol{w} = 1,
\end{aligned}
\tag{4}
$$

where $\boldsymbol{w} = (w_1, w_2, \cdots, w_M)^T$. After simplification, we get the following TSCCA model:

$$
\begin{aligned}
\underset{\boldsymbol{u},\boldsymbol{v},\boldsymbol{w}}{\text{maximize}} \quad & -\mathcal{A} \bar{\times}_1 \boldsymbol{u} \bar{\times}_2 \boldsymbol{v} \bar{\times}_3 \boldsymbol{w} \\
\text{subject to} \quad & \|\boldsymbol{u}\|_0 \le k_u, \|\boldsymbol{v}\|_0 \le k_v, \|\boldsymbol{w}\|_0 \le k_w, \\
& \boldsymbol{u}^T \boldsymbol{u} = 1, \boldsymbol{v}^T \boldsymbol{v} = 1, \boldsymbol{w}^T \boldsymbol{w} = 1, \\
\text{with} \quad & \mathcal{A}_{::i} = \boldsymbol{X}^{i^T} \boldsymbol{Y}^i \, i = 1, \cdots, M,
\end{aligned}
\tag{5}
$$

where $\mathcal{A} \bar{\times}_i \boldsymbol{z}$ ($i = 1, 2, 3$) denotes the $i$-mode (vector) product of a tensor $\mathcal{A} \in \mathbb{R}^{I_1 \times I_2 \times I_3}$ with a column $\boldsymbol{z} \in \mathbb{R}^{I_i}$, and $\mathcal{A}_{::i}$ is frontal slice and also written as $\mathcal{A}_i$. More detailed definitions about tensor operations can be found in [38].

## Proposed optimization algorithm

Recently, a global block-coordinate update algorithm has been proposed to solve a class of nonconvex optimization problems [39]. The block-coordinate descent algorithm is also called as alternating iteration algorithm which updates one factor at a time with the others fixed. Inspired by the algorithm, we develop a block-coordinate descent algorithm to solve the above problem (5):

$$
\begin{aligned}
\boldsymbol{u}^{k+1} \quad &\leftarrow \quad \underset{\boldsymbol{u}^T \boldsymbol{u} = 1, \|\boldsymbol{u}\|_0 \le k_u}{\arg\min} \; f(\boldsymbol{u}, \boldsymbol{v}^k, \boldsymbol{w}^k), \\
\boldsymbol{v}^{k+1} \quad &\leftarrow \quad \underset{\boldsymbol{v}^T \boldsymbol{v} = 1, \|\boldsymbol{v}\|_0 \le k_v}{\arg\min} \; f(\boldsymbol{u}^{k+1}, \boldsymbol{v}, \boldsymbol{w}^k), \\
\boldsymbol{w}^{k+1} \quad &\leftarrow \quad \underset{\boldsymbol{w}^T \boldsymbol{w} = 1, \|\boldsymbol{w}\|_0 \le k_w}{\arg\min} \; f(\boldsymbol{u}^{k+1}, \boldsymbol{v}^{k+1}, \boldsymbol{w}),
\end{aligned}
\tag{6}
$$

where $f(\boldsymbol{u}, \boldsymbol{v}, \boldsymbol{w}) = -\mathcal{A} \bar{\times}_1 \boldsymbol{u} \bar{\times}_2 \boldsymbol{v} \bar{\times}_3 \boldsymbol{w}$. To implement it, we need to solve three sub-problems in Eq (6). Taking the first as an example, with $\boldsymbol{v}$ and $\boldsymbol{w}$ fixed, it is equivalent to solve

$$
\begin{aligned}
\underset{\boldsymbol{u}}{\text{minimize}} \quad & -\boldsymbol{u}^T \boldsymbol{z} \\
\text{subject to} \quad & \boldsymbol{u}^T \boldsymbol{u} = 1, \|\boldsymbol{u}\|_0 \le k,
\end{aligned}
\tag{7}
$$

where $\boldsymbol{z} = \mathcal{A} \bar{\times}_2 \boldsymbol{v} \bar{\times}_3 \boldsymbol{w}$. For convenience, we define a $k$-sparse projection operator $\Pi(\cdot, k)$ for a given $\boldsymbol{z} \in \mathbb{R}^p$ with $k \le p$:

$$
[\Pi(z, k)]_i = \begin{cases} z_i, & \text{if } i \in \text{support}(z, k) \\ 0, & \text{otherwise} \end{cases}
\tag{8}
$$

where $support(\boldsymbol{z}, k)$ is a set of indices of $\boldsymbol{z}$ with the largest $k$ absolute values. For example, if $z = (-6, 4, 5, 2, -1, 3)^T$, then $\Pi(z, 3) = (-6, 4, 5, 0, 0, 0)^T$. We have Proposition 1 to solve Eq (7) and its proof is detailed in S1 Text.

**Proposition 1**. *Suppose $\boldsymbol{z}$ is a non-zero vector, then the solution of problem (7) is* $\boldsymbol{u}^* = \frac{\Pi(z,k)}{\|\Pi(z,k)\|_2}$.

Based on Proposition 1, we develop a block-coordinate descent algorithm to solve (5). The details of this algorithm is shown in Algorithm 1 and its stopping condition, convergence analysis and computational complexity are given in S1 Text.

**Algorithm 1** TSCCA algorithm solves Eq (5)

**Require:** $\boldsymbol{X}^i \in \mathbb{R}^{n_i \times p}$ (gene expression data) and $\boldsymbol{Y}^i \in \mathbb{R}^{n_i \times q}$ (miRNA expression data) for $i = 1, \cdots, M$ (cancer types); Parameters: $k_u$, $k_v$, and $k_w$.

**Ensure:** $\boldsymbol{u}$, $\boldsymbol{v}$, $\boldsymbol{w}$ and singular value $d$.

1: Compute $\mathcal{A}_i = (\boldsymbol{X}^i)^T \boldsymbol{Y}^i$, $i = 1, \ldots, M$

2: Initialize $\boldsymbol{w} = \left(\frac{1}{\sqrt{M}}, \ldots, \frac{1}{\sqrt{M}}\right)^T$ with $\|\boldsymbol{w}\| = 1$

3: Initialize $\boldsymbol{u}$, $\boldsymbol{v}$ using the principal left and right singular vectors of $\sum_{i=1}^{M} w_i \mathcal{A}_i$

4: **repeat**

5:     Compute a matrix $\boldsymbol{C} = \sum_{i=1}^{M} w_i \mathcal{A}_i$

6: Let $\boldsymbol{z}_u = \boldsymbol{C} \boldsymbol{v}$

7: $\boldsymbol{u} \leftarrow \frac{\Pi(z_u, k_u)}{\|\Pi(z_u, k_u)\|_2}$

8:     Let $\boldsymbol{z}_v = \boldsymbol{C}^T \boldsymbol{u}$

9:     $\boldsymbol{v} \leftarrow \frac{\Pi(z_v, k_v)}{\|\Pi(z_v, k_v)\|_2}$

10:     Let $\boldsymbol{z}_w = [\boldsymbol{u}^T \mathcal{A}_1 \boldsymbol{v}, \ldots, \boldsymbol{u}^T \mathcal{A}_M \boldsymbol{v}]^T$

11:     $\boldsymbol{w} \leftarrow \frac{\Pi(z_w, k_w)}{\|\Pi(z_w, k_w)\|_2}$

12: **until** convergence of $\boldsymbol{u}$, $\boldsymbol{v}$ and $\boldsymbol{w}$

13: $d = \mathcal{A} \bar{\times}_1 \boldsymbol{u} \bar{\times}_2 \boldsymbol{v} \bar{\times}_3 \boldsymbol{w}$

14: **return** $\boldsymbol{u}$, $\boldsymbol{v}$, $\boldsymbol{w}$ and singular value $d$

## Determination of cancer-miRNA-gene modules

Based on the output of Algorithm 1, the non-zero genes in $\boldsymbol{u}$, the non-zero miRNAs in $\boldsymbol{v}$ and the non-zero cancer types in $\boldsymbol{w}$ together are considered as a cancer-miRNA-gene functional module (Fig 1C). Furthermore, we also extend Algorithm 1 to identify the next module by updating the input $\mathcal{A} := \mathcal{A} - d \cdot \boldsymbol{u} \circ \boldsymbol{v} \circ \boldsymbol{w}$, where $d = \mathcal{A} \bar{\times}_1 \boldsymbol{u} \bar{\times}_2 \boldsymbol{v} \bar{\times}_3 \boldsymbol{w}$ and it is also called singular value, reflecting the relative importance of a corresponding module (See Algorithm 2 in S1 Text). We carefully discuss the parameter selection issue of Algorithm 1 (See S1 Text for more detail).

## Modularity

For a given module with a gene set $I$, a miRNA set $J$ and a cancer type set $K$, we define a modularity score:

$$\text{Modularity} = \frac{1}{|I||J||K|} \sum_{i \in I, j \in J, k \in K} |C_{ijk}|, \tag{9}$$

where $C_{ijk}$ is a Pearson correlation coefficient (PCC) between gene $i$ and miRNA $j$ in the cancer $k$. A high modularity score indicates that these genes and miRNAs within the module are strongly co-expressed across these selected cancers within the module.

## Results

### Application to the TCGA data

We applied TSCCA to matched miRNA and gene expression data from TCGA consisting of 9645 cancer patients across 33 cancer types (Fig 2A and S2 Table). All output of TSCCA is detailed in S3–S6 Tables. We discovered 50 cancer-gene-miRNA modules (S7 Table). Each identified module contains about 100 genes, 10 miRNAs and 20 cancer types. Regarding the characteristics of TSCCA when it was applied to the TCGA data, we observed that (1) TSCCA converged in about 20 steps (S1 Fig) and it took a total of about 1 hour on a personal laptop. (2) The modularity scores of these modules have a strong correlation with their corresponding singular values of TSCCA model (PCC $r = 0.98$ with $P < 0.001$, Fig 2B). In addition, we also used permutation test to assess the number of overlapping elements between any two modules (S8 Table, see section 7 in S1 Text for more detail). Only 51 out of 1225 pairs of module from these identified modules are significantly overlapping with permutation test $P < 0.05$, indicating that these identified modules are statistically independent patterns.

### Statistical analysis of correlation of modules

To evaluate the correlations between genes and miRNAs within each module, we randomly generated 1,000 modules with the same size as these identified modules. The identified modules with $P$-values smaller than 0.05/50 were considered as significant ones. We found that the modularity scores of all modules are significantly larger than those of the random ones (Fig 2C and S1 Text). For each cancer type on the TCGA data, we also computed a basic modularity score based on all considered miRNAs ($n = 523$) and genes ($n = 7889$). We observed that 33 basic modularity scores of TCGA 33 cancer types are distributed between about $0.1 \sim 0.2$ (S9 Table). For example, the basic modularity score of TGCT is the largest with Modularity = 0.21 and CGA is the smallest with Modularity = 0.086. Full details on these 33 cancer types are given in S9 Table. We observed that the modularity scores of these identified modules are far greater than the corresponding basic modularity score in these selected cancers.

### Module miRNAs and genes are strongly implicated in cancer

To assess whether these identified modules are related to cancer, we first collected a total of 1793 genes and 122 miRNAs via combining all the modules. In addition, we also collected a cancer gene set from the allOnco database and a cancer miRNA set from [29]. As we expected, we found that 328 out of 1793 genes are cancer genes (Hypergeometric test $P$ = 1.47e-06) (Fig 2D), and 73 out of 122 miRNAs are cancer miRNAs (Hypergeometric test $P$ = 3.38e-03) (Fig 2E). In addition, we also used hypergeometric test to evaluate whether the number of cancer genes or cancer miRNAs within each identified module is significantly larger than expected by

chance (S10 Table and S1 Text). We found that each module contains an average of 6 cancer miRNAs and 20 cancer genes. There are 8 out of 50 modules including significantly more cancer miRNAs and 15 out of 50 modules including significantly more cancer genes. For example, module 1 contains 31 cancer genes (fold enrichment = 2.1, hypergeometric test $P < 0.05$) and module 4 contains 20 cancer genes (fold enrichment = 1.7, hypergeometric test $P < 0.05$).

### Characteristics of modules in different cancers

To visualize the co-expressed pattern of each identified cancer-miRNA-gene module, we first calculated a Pearson correlation matrix between the genes and the miRNAs within the module based on the corresponding miRNA and mRNA expression data for each cancer within the module. We then drew a heatmap to show the co-expressed pattern using these correlation matrices. The heatmaps of these identified modules are given in S2 Fig. We found that some identified modules show different co-expressed patterns in different cancer types. For example, the genes and the miRNAs within module 1 show strong positive correlation on all selected cancers (Fig 3A), those within module 2 are both positively and negatively correlated on all selected cancers (Fig 3B), whereas those within module 5 show strong negative correlation on all selected cancers (Fig 3C). These results suggest that miRNA-gene regulation in cancer are very complex.

We further investigated whether these modules are specifically related with some cancer types by visualizing the matrix **W** (Fig 4). **W** is the output matrix of Algorithm 2 (See section

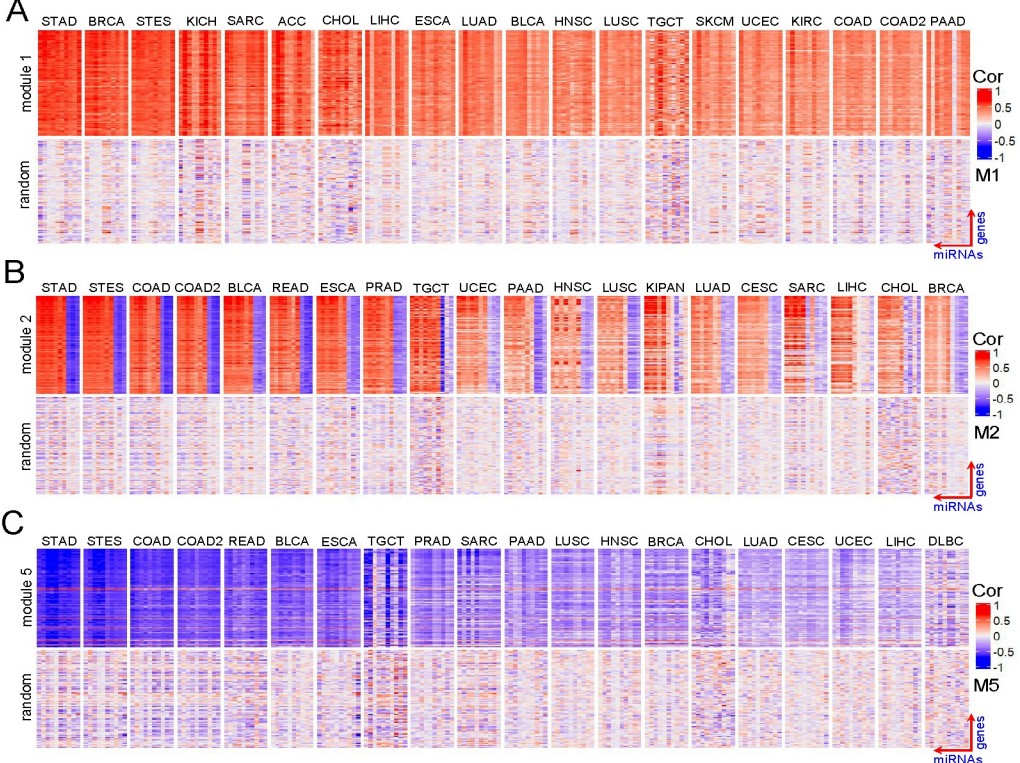

**Fig 3. Heatmap of cancer-miRNA-gene modules identified by TSCCA in the TCGA dataset.** The top half of (A) corresponds to the module 1 (row corresponds to gene, column corresponds to miRNA) and the lower part of (A) is a random module for comparison. Similar setting is used for module 2 and module 5 in (B) and (C) respectively. (A), (B) and (C) show three different co-expression patterns.

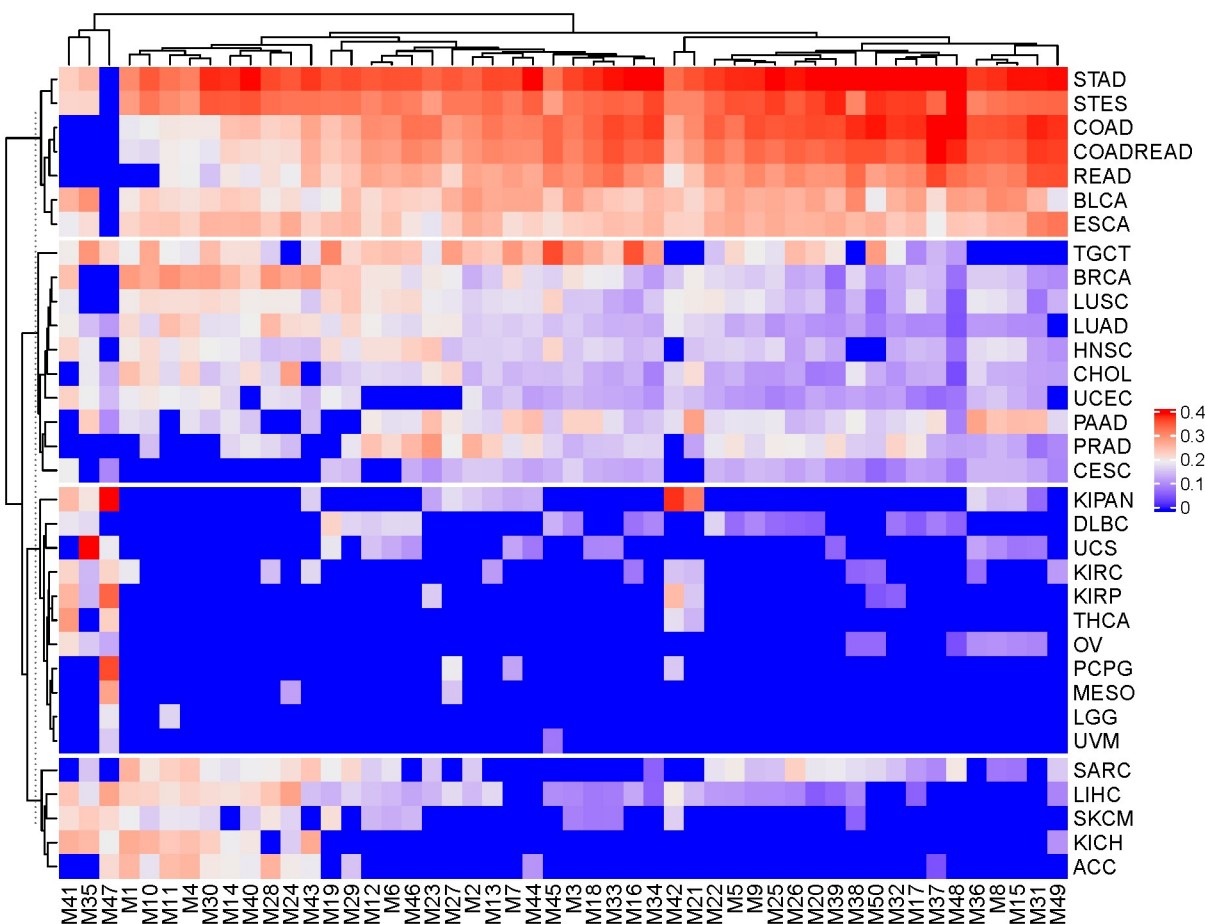

**Fig 4. Heatmap showing *W*, which is the output matrix of Algorithm 2 (See S1 Text), when it was applied to the TCGA data.** Each column corresponds to a module and each row corresponds to a cancer type and |$W_{ij}$| reflects the co-expressed intensity of between the genes and the miRNAs within the module $j$ on the cancer $i$. A hierarchical clustering method was used to cluster the rows (cancer types) into four clusters.

6 in S1 Text), whose each column corresponds to a module, and each row corresponds to a cancer type. The absolute value of $W_{ij}$ reflects the co-expressed intensity of between the genes and the miRNAs within the module $j$ on the cancer $i$. We first observed that there are only three negative elements in *W* (S3(A) and S3(B) Fig), i.e., (Module 31, TGCT) is −0.145, (Module 49, TGCT) is −0.23, and (Module 49, UCS) is −0.138. Interestingly, we also observed that the miRNAs and genes within module 31 are positively correlated in TGCT cancer type, but are negatively correlated in other cancer types, and module 49 are positively correlated in TGCT and UCS cancer types, but are negatively correlated in other cancer types (S3(C) Fig). In addition, a hierarchical clustering method was used to cluster the rows (cancer types) of *W* and the 33 cancer types were divided into 4 clusters. The first cluster (including STAD, STES, COAD, COADREAD, READ, BLCA and ESCA) has the strongest weighted values of *W*. The second cluster contains TGCT, BRCA, LUSC, LUAD, HNSC, CHOL, UCEC, PAAD, PRAD and CESC, where the LUSC and LUAD show very similar patterns in different modules. The third cluster (including KIPAN, DLBC, UCS, KIRC, KIRP, THCA, OV, PCPG, MESO, LGG and UVM) has the weakest weighted values. Several cancer types in the third cluster show module-specific characteristics. For example, UVM is specifically related with module 45, and LGG is specifically related with module 11. Importantly,

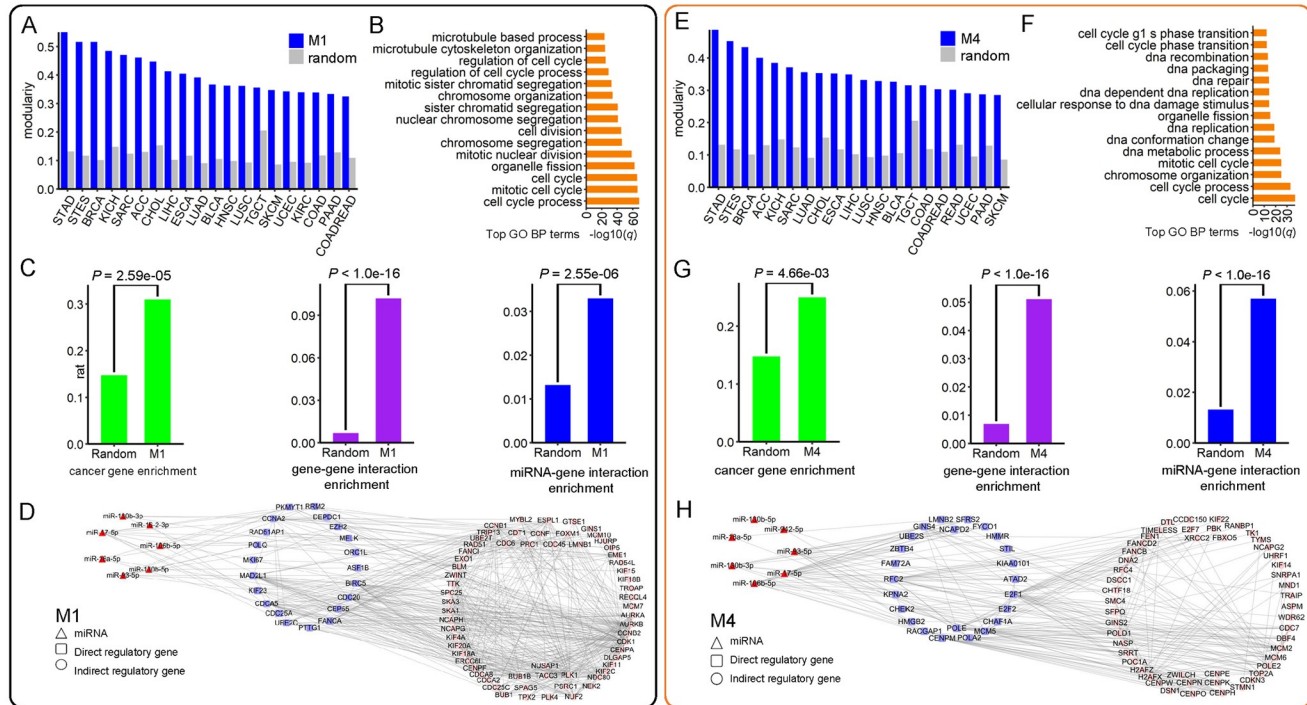

**Fig 5. Illustration of two cancer-miRNA-gene modules identified by TSCCA in the TCGA dataset.** The results on module 1 are shown in (A), (B), (C) and (D), while the results on module 4 are shown in (E), (F), (G) and (H). (A) Bar plot showing modularity scores of module 1 and a random one for different cancer types. (B) Top enriched GO BP terms on the genes within module 1. (C) Cancer gene enrichment, gene-gene interaction enrichment and miRNA-gene interaction enrichment of module 1 and the corresponding *P*-values were computed using the right-tailed hypergeometric test. (D) Largest connected miRNA-gene subnetwork of module 1 (including 7 miRNAs and 84 genes and 538 edges), where the miRNAs directly regulate 21 genes and the 21 genes regulate 63 other genes. Similar setting was used for module 4 in (E), (F), (G) and (H). (H) Largest connected miRNA-gene subnetwork of module 4 (including 7 miRNAs and 75 genes and 309 edges), where the miRNAs directly regulate 24 genes and the 24 genes regulate 51 other genes.

the results of the following survival analysis also show that module 11 is the most important and clinically relevant module with LGG in all the modules. The fourth cluster contains SARC, LIHC, SKCM, KICH and ACC. We note that TSCCA is an explorative tool, which identifies the "strongest" modular patterns in the current multiple cancer data. This means that in a subset of cancer data, it could identify other significant modules. For example, most of the 50 modules identified by TSCCA on the TCGA dataset are enriched in 60% of cancers, while other cancers are rare. To this end, we may extract a subset of cancers from the cluster 3 in Fig 4 and then re-use TSCCA to extract some modules on a subset of the previous data (across 18 cancers). We found some new modules with significant modularity scores, and more details are given in S4 Fig. This procedure will overcome the limit that a small number of cancers may dominate the results for TSCCA.

We also calculated a modularity score for each cancer type of an identified module. Two examples are shown in Fig 5A and 5E. These modularity scores of different cancers for the two examples are larger than those of the random ones. All the results suggest that the miRNAs and the genes are strongly co-expressed on these selected cancers for each module.

## Cooperativity of genes and miRNAs within modules

To evaluate the biological relevance of the modules, we performed GOBP, KEGG and Reactome pathway enrichment analysis for the genes within each module (See section 13 in S1 Text). We downloaded the gene functional annotations including GOBP, KEGG and reactome

pathways from MSigDB [30]. We found that 84% (42 out of 50) modules identified by TSCCA are significantly related with at least a functional term with a benjamini-hochberg (BH) adjusted $P < 0.05$ (S11 Table) and different modules tend to be enriched in different terms. On average, each module is significantly enriched in 40 GOBP terms (S12 Table), 2 KEGG terms (S13 Table) and 8 Reactome terms (S14 Table). For example, the top enriched GOBP terms of module 1 includes cell cycle process, mitotic cell cycle, cell cycle, etc (Fig 5B), and the top enriched GOBP terms of module 4 includes cell cycle, cell cycle process and chromosome organization (Fig 5F). Importantly, cell cycle process has been reported to be one of 10 oncogenic signaling pathway [40].

To assess whether the genes within module tend to be densely connected on the gene interaction network, we computed the numbers of gene interactions from this network for each module (S10 Table). We found that 64% (32 out of 50) modules contain significantly more gene interactions than expected by chance (Hypergeometric test $P < 0.001$). For example, module 1 contains 505 gene interactions with 15-fold enrichment of the interaction density of the gene interaction network (Hypergeometric test $P < 1.0e$-16, Fig 5C middle), and module 4 contains 253 gene-gene interactions with 7-fold enrichment (Hypergeometric test $P < 1.0e$-16, Fig 5G middle). In addition, to avoid the influence of degree in the gene interaction network, we developed a statistical permutation test method to perform the gene-gene interaction set enrichment, and found that 88% (44 out of 50) modules contain significantly more gene interactions than expected by chance (Permutation test $P < 0.05$, see section 23 in S1 text). All the above results suggest that the genes within each identified module tend to cooperate with each other.

Previous studies have shown that miRNAs co-regulate gene expression in a cooperative form and participate in cellular activities [5]. So, we expect the miRNAs within module to be cooperative. To this end, we collected a miRNA family data set from miRbase database [9]. We found that 92% (46 out of 50) modules have at least two miRNAs in the same family (Permutation test $P < 0.01$, see section 17 in S1 Text and S15 Table). For example, the members of module 1 including hsa-miR-17–5p, hsa-miR-18a-5p, hsa-miR-93–5p, hsa-miR-106b-5p, and hsa-miR-106b-3p belong to miR-17 family, which has been reported to be associated with cancer [41]. Module 8 includes seven miRNAs, which are hsa-miR-200b-5p, hsa-miR-200b-3p, hsa-miR-200c-5p, hsa-miR-200c-3p, hsa-miR-200a-5p, hsa-miR-200a-3p, and hsa-miR-429 and they belong to miR-8 family, which has been reported to be associated with cancer [42].

We also evaluated the cooperation of the genes and the miRNAs within module from statistical significance using a permutation test method. To this end, we computed the average of gene-gene/miRNA-miRNA absolute PCCs of any two genes/miRNAs within a given module (denoted as gene/miRNA modularity). We found that the gene/miRNA modularity scores of all the identified modules are significantly larger than those of 1000 modules randomly generated (Permutation test $P < 0.01$) (Fig 6A and 6B). On average, the miRNA modularity score is about 0.5, and gene modularity is about 0.45 for these identified modules. These results demonstrate that the genes/miRNAs within a module tend to cooperate from the perspective of co-expression.

## miRNA-gene regulatory network analysis of modules

To evaluate whether the regulatory relationship between miRNAs and genes within a given module tends to be verified experimentally, we computed the number of experimentally validated miRNA-gene interactions between these miRNAs and genes within the module. These experimentally validated interactions are from a miRNA-gene interaction network, which is collected from the miRTarBase database [27]. We found that 38% (19 out of 50) modules

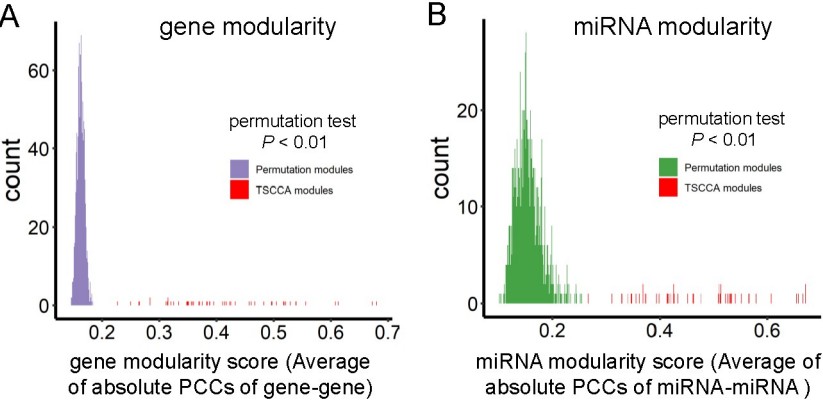

**Fig 6. Statistical analysis of PCCs of module miRNAs/genes using permutation test.** (A) The average of absolute gene-gene PCCs of the genes within each module (Permutation test $P < 0.01$). (B) The same results about miRNAs.

contain the number of validated miRNA-gene interactions are significantly more than expected by chance (Hypergeometric test $P < 0.05$) (S10 Table). For example, module 1 contains 33 validated miRNA-gene interactions with 2.5-fold enrichment of the whole experimentally validated miRNA-gene network (Fig 5C right), and module 4 contains 57 validated miRNA-gene interactions (Fig 5G right). In addition, to avoid the influence of degree for miRNAs in the miRNA-gene network, we developed a statistical permutation test method to perform the miRNA-gene interaction set enrichment (See section 23 in S1 Text). There are 28% (14 out of 50) modules which contain significantly more miRNA-gene interactions than expected by chance (Permutation test $P < 0.05$).

For each identified miRNA-gene module, we have confirmed that some miRNA-gene interactions are verified by the miRTarBase database, while there are also many miRNA-gene pairs are not verified by the database. Furthermore, based on the experimentally validated miRNA-gene and gene-gene interactions, we built a three-layer miRNA-gene regulatory network for each module: miRNAs regulate genes and these genes regulate the other genes within the three-layer network (S5(A) Fig). We found that 70% modules have at least three miRNAs participating in a three-layer network (Permutation test $P < 0.01$, see section 17 in S1 Text). The detailed results are shown in S16 Table. For example, we extracted a largest connected miRNA-gene subnetwork of module 1 (including 7 miRNAs, 84 genes and 538 edges), where the miRNAs directly regulate 21 genes and the 21 genes regulate 63 other genes (Fig 5D), and a largest connected miRNA-gene subnetwork of module 4 (including 7 miRNAs, 75 genes and 309 edges), where the miRNAs directly regulate 24 genes and the 24 genes regulate 51 other genes (Fig 5H). Interestingly, we also collected a total of 1793 genes and 122 miRNAs via combining all identified modules and found 3619 experimentally validated miRNA-gene interactions with hypergeometric test $P = 3.5e-43$ (S5(B) Fig).

## Survival analysis of modules

To evaluate whether the identified modules can be seen as prognostic biomarkers, we further investigated the association between the expression of both miRNAs and genes within the module and survival time. For each module and for each cancer within the module, we first extracted the first principal component (PC1) based on the expression data of these genes and miRNAs within the module. We then divided the cancer samples into two groups based the median value of the PC1 and log-rank test was used to assess the difference between the two groups of samples and a $P$-value was computed. All computed $P$-values were corrected

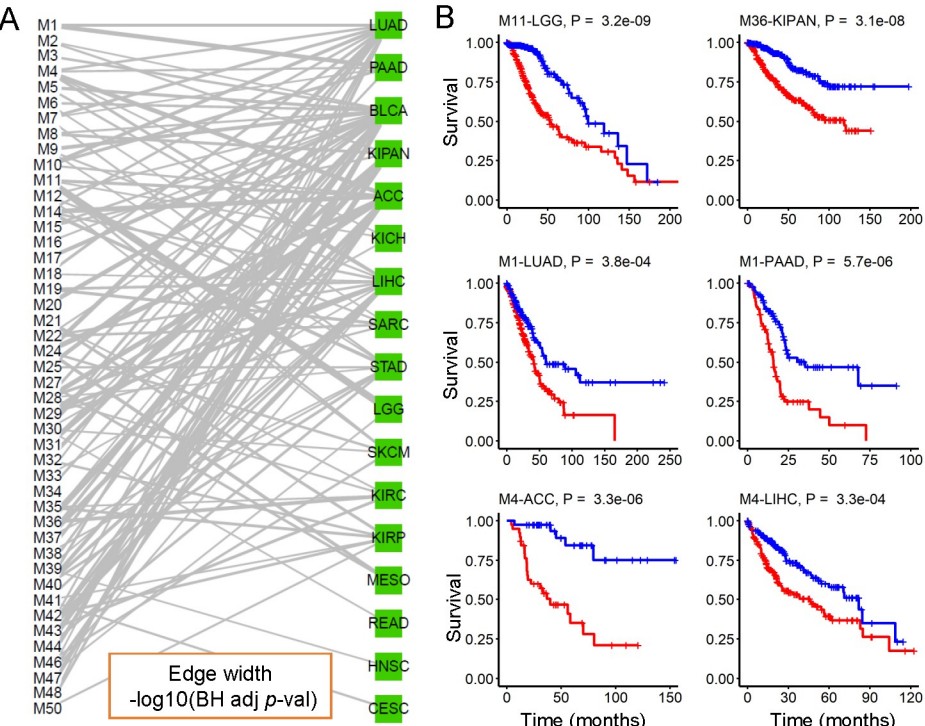

**Fig 7. Survival analysis of modules.** (A) showing a bipartite graph between the identified modules and the different cancer types based on these -log₁₀(BH adjusted *P*-value). For each identified module and each cancer within the module, we first extracted the first principal component (PC1) based on the expressed matrix of both miRNAs and genes within the module from the cancer type. We then divided the samples from the cancer type into two groups based on the median value of PC1 and a *P*-value was compute using log-rank test. In the graph, we only kept these edges/relationships between the modules and cancer types with adjusted *P* < 0.05. (B) Some cancer-miRNA-gene modules relate to survival time. For a given cancer type and a given module, the Kaplan-Meier survival curves were drawn for each group, and "+" denotes the censoring patient. Each sub-figure corresponds to a module and a cancer type. For example, Module 11 has a significant *P* = 3.2e-09 for LGG (cancer type), written as "M11-LGG, *P* = 3.2e-09".

using the BH adjusted method. Based on these -log10(BH adjusted *P*-value) scores, we built a bipartite graph between the modules and the cancer types (Fig 7A and S17 Table). In the bipartite graph, we only kept these edges between the modules and the cancer types with BH adjusted *P* < 0.05. In total, there are 45 modules, 17 cancer types and 116 significant module-cancer edges in the bipartite graph. We found that 80% modules are significantly related to the survival time on at least one cancer. For example, we found that M11-LGG and M36-KIPAN edges have the largest weight value (i.e., smallest *P*-value) in the bipartite graph. Module 11 is the most important and clinically relevant module to LGG (Log-rank test *P* = 3.18e-06) and module 36 is a clinically relevant module to KIPAN (Log-rank test *P* = 1.53e-05) (Fig 7B).

We also considered the expression of each miRNA within a module as the prognostic scores (S18 Table). On average, we found that two clinically relevant miRNAs with BH adjusted *P* < 0.05 for each cancer. Some important and clinically relevant miRNAs were found. For example, the two most significant miRNAs are hsa-miR-15b-3p of module 11 for LGG with log-rank test *P* = 3.33e-06, and hsa-miR-130b-3p of module 43 for KIPAN with log-rank test *P* = 1.13e-05. In addition, three miRNAs (hsa-miR-93–3p, hsa-miR-130b-5p and hsa-miR-130b-3p) of module 19 correlate with survival in ACC and four miRNAs (hsa-let-7c-5p, hsa-miR-99a-5p, hsa-miR-125b-5p and hsa-miR-125b-2–3p) of module 3 correlate with survival

in BLCA. These results reveal that some modules can be used as prognostic biomarkers in multiple cancer types.

## Case studies

Based on the above functional analysis, we found that some identified modules show diverse biological functions and relevance from different views (S19 Table). We took modules 1, 4, and 11 as examples. The module 1 consists of 100 genes, 10 miRNAs and 20 cancers, of which 5 cancer miRNAs and 31 cancer genes (Hypergeometric test $P$ = 2.59e-05). The correlations between miRNAs and genes across the selected cancer types are statistically significant compared to random ones (Permutation test $P < 0.001$). For five cancer types (including STAD, BRCA, STES, KICH and SARC), the expression pattern of miRNAs and genes within the module is significantly related with their patient survival respectively (Log-rank test BH adjusted $P < 0.05$, see section 18 in S1 Text). Therefore, we may consider module 1 as a potential prognostic biomarker for these five cancer types. Moreover, the module genes are enriched with a large number of cancer-related functional terms including GOBP terms (cell cycle process, mitotic cell cycle, cell cycle, chromosome segregation and cell division) and KEGG pathways (cell cycle, oocyte meiosis, progesterone mediated oocyte maturation, homologous recombination and p53 signaling pathway), suggesting its strong cancer relevance. Recent studies have shown that these cell cycle-related functions are related to multiple cancer processes [40, 43]. On the other hand, five module miRNAs (hsa-miR-17–5p, hsa-miR-18a-5p, hsa-miR-93–5p, hsa-miR-106b-5p and hsa-miR-106b-3p) belong to miR-17 family, which has been reported to be related to cancer [41, 44]. Finally, we also found that 6 of 10 miRNAs is related with patient survival in at least one cancer type (Log-rank test BH adjusted $P < 0.05$) (S18 Table). For example, the expression of hsa-miR-130b-5p and hsa-miR-130b-3p are significantly related with ACC patient survival.

The module 4 contains 5 cancer miRNAs and 25 cancer genes (Hypergeometric test $P$ = 4.66e-03). The correlations between miRNAs and genes across the selected cancer types within this module are statistically significant compared to random ones (Permutation test $P < 0.001$). This module is significantly related to the survival time in five cancer types (ACC, LIHC, LUAD, PAAD, KICH). The genes within the module are enriched with some cancer-related functional terms including GOBP terms (cell cycle, cell cycle process, chromosome organization, mitotic cell cycle and DNA metabolic process) and KEGG pathways (DNA replication, base excision repair, nucleotide excision repair, cell cycle, pyrimidine metabolism). Boyer *et al.* have reported that DNA replication pathway plays an important role in cancer [45]. More importantly, we found that 57 miRNA-gene interactions between the miRNAs and genes within this module were verified before. Collecting the gene-gene network from PPI network, we construct a miRNA-gene-gene regulatory sub-network where there are 7 miRNAs, 75 genes and 309 edges (Fig 5H and S16 Table).

The last example, module 11 exhibits distinct biological relevance with LGG (Brain Lower Grade Glioma) in terms of miRNAs and genes. Firstly, the miRNAs and genes across the selected cancer types within the module show strong correlations (Permutation test $P < 0.001$). Secondly, the genes within this module are enriched with several cancer-related KEGG pathways including cell cycle, small cell lung cancer, DNA replication, mismatch repair. As mentioned earlier, cell cycle and DNA replication pathways have been reported to play an important role in cancer. Thirdly, 36 miRNA-gene interactions between the miRNAs and genes within this module were verified by miRTarBase database. We also construct a miRNA-gene-gene regulatory sub-network, which contains 7 miRNAs, 68 genes and 208 miRNA-gene edges (S16 Table). Importantly, two miRNAs (hsa-miR-130b-5p and hsa-miR-130b-3p) within

the module belong to mir-130 family, which have been reported as potential biomarkers for brain cancer [46–48]. Especially, the expression pattern of miRNAs and genes within the module is significantly related with LGG patient survival (Log-rank test BH adjusted $P$ = 3.18e-06).

### Comparison on the simulated data

In this section, we compared TSCCA with SCCA [35] and Modularity_SA on a set of simulated data. Modularity_SA is a modularity-based simulated annealing (Modularity_SA) method (See section 19 in S1 Text), which uses a simulated annealing algorithm to maximize the modularity index (Eq 9) for extracting a cancer-miRNA-gene module.

We generated a synthetic miRNA-gene correlation tensor $\mathcal{A} \in \mathbb{R}^{300 \times 30 \times 4}$ with 300 genes and 30 miRNAs and 4 cancers, where (1) $\mathcal{A}_1[i,j] \sim N(0.5, 0.2^2)$ when $1 \leq i \leq 100$ and $1 \leq j \leq 10$, and $\mathcal{A}_1[i,j] \sim N(-0.5, 0.2^2)$ when $101 \leq i \leq 200$ and $11 \leq j \leq 20$, and the other elements are from $N(0, 0.2^2)$; (2) $\mathcal{A}_2[i,j] \sim N(-0.5, 0.2^2)$ when $1 \leq i \leq 100$ and $1 \leq j \leq 10$, and $\mathcal{A}_2[i,j] \sim N(0.5, 0.2^2)$ when $201 \leq i \leq 300$ and $21 \leq j \leq 30$, and the other elements are from $N(0, 0.2^2)$; (3) $\mathcal{A}_3[i,j] \sim N(0.5, 0.2^2)$ when $101 \leq i \leq 200$ and $11 \leq j \leq 20$, and $\mathcal{A}_3[i,j] \sim N(-0.5, 0.2^2)$ when $201 \leq i \leq 300$ and $21 \leq j \leq 30$, and the other elements are from $N(0, 0.2^2)$; (4) $\mathcal{A}_4[i,j] \sim N(0, 0.2^2)$ for any $i$ and $j$. We repeatedly generated 50 tensors ($\mathcal{A}$s) and Fig 8A shows an $\mathcal{A}$. For each $\mathcal{A}$, we applied SCCA to each single miRNA-gene correlation

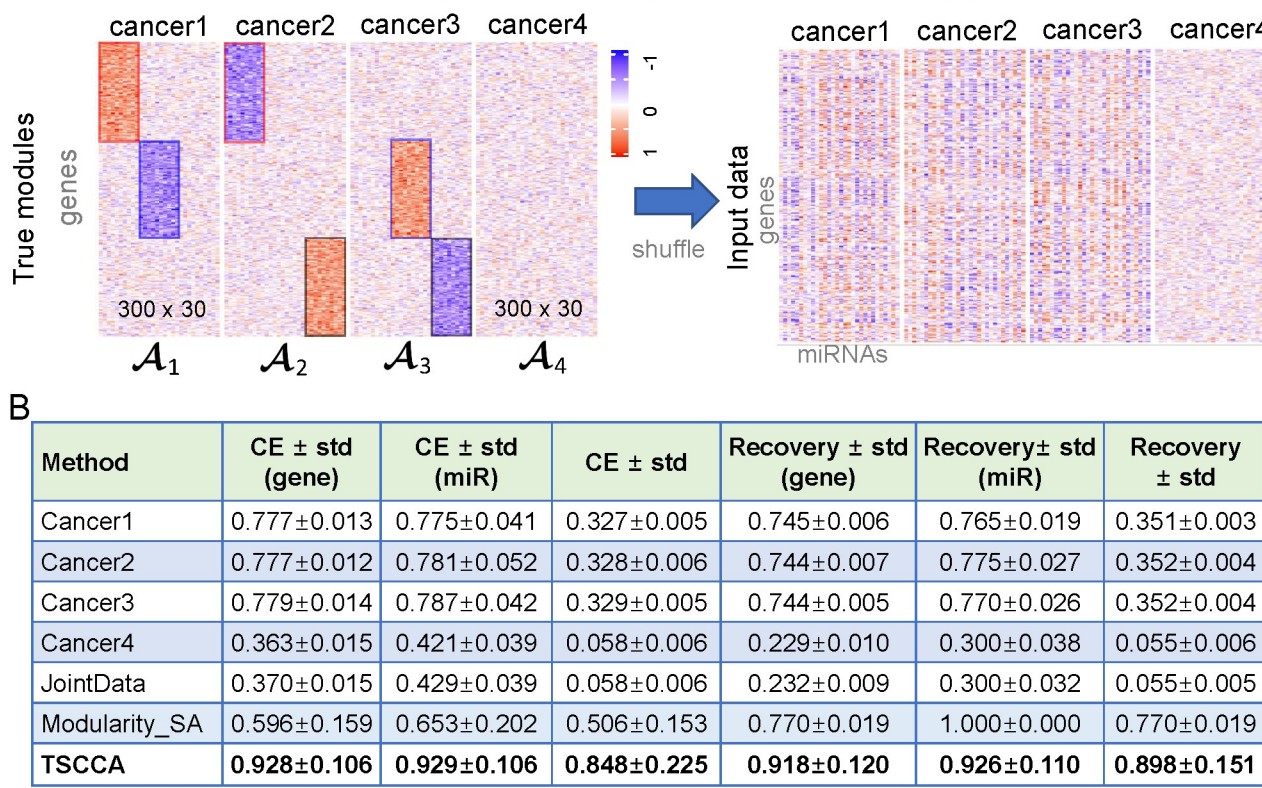

**Fig 8. Comparison of results from different algorithms on the simulated data and TCGA data.** (A) A synthetic miRNA-gene correlation tensor $\mathcal{A}$, which contains four matrices with the same number of genes (rows) and miRNAs (columns), and includes three true modules framed by rectangular boxes of different colors. The shuffled $\mathcal{A}$ is as the input of tested methods by shuffling the genes (rows) and miRNAs (columns) of $\mathcal{A}$. (B) Comparison of different methods in terms of CE ± std and Recovery ± std on the simulated data. The Recovery and CE scores are computed based on $\mathcal{A}$s generated repeatedly.

matrix $\mathcal{A}_i$ and the joint data defined as $\sum_i^4 \mathcal{A}_i$. To ensure fairness of comparison between TSCCA, SCCA and Modularity_SA, their parameters are consistent with the size of true modules. We assessed the similarity between the true modules and the prediction modules through the use of two metrics: Clustering error (CE) score and Recovery score (S20 and S21 Tables, see section 20 in S1 Text for more detail). The results show that TSCCA is superior to other methods in terms of Recovery and CE scores (Fig 8B). More results and description on the simulated data with different variances are given in S22 and S23 Tables (See section 21 in S1 Text for more detail). We found that SCCA has two disadvantages on the single cancer simulated data: (1) SCCA always loses a real module on the simulated data. For example, SCCA misses the module 3 when it was applied to $\mathcal{A}_1$, and misses the module 2 when it was applied to $\mathcal{A}_2$, and SCCA misses all modules when was applied to the noise matrix $\mathcal{A}_4$. (2) SCCA cannot make feature selection about the cancer types, i.e., SCCA cannot assess the importance of the module for different cancers. Additionally, Modularity_SA has two short-comings: (1) it misses some real members of the true modules; and (2) it is more time-con-suming compared to TSCCA.

## Comparison on the TCGA data

In this section, we compared TSCCA with SCCA and multiple tri-clustering methods on the TCGA data. Firstly, we used SCCA to identify 50 modules on each cancer data set and com-pared TSCCA with SCCA in terms of modularity scores and multiple biological indicators (S24 Table). The parameters of SCCA is consistent with the parameters of TSCCA with $k_u = 200$, $k_v = 10$ and $k_w = 20$ when applying to the TCGA data. For a single cancer data, SCCA also ensures that the expression of miRNAs and genes within the identified modules are correlated in the specific cancer data (See the eighth column in S24 Table), but it failed to ensure that the miRNAs and genes with the identified modules are correlated in most cancer types (See the seventh column in S24 Table). Thus, TSCCA is more suitable to multi-cancer data compared to SCCA.

Secondly, we also compared TSCCA with multiple tri-clustering methods including Modu-larity_SA and Sparse Canonical Polyadic decomposition (SCP) which uses $\ell_1$-regularization to force sparse [49], and two merit-function based methods including "Variance" (See Eq 1 in [50]) and "Mean squared residue (MSR)" (See Eq 3 in [50]). The two merit-functions are optimized by using annealing algorithm. Var_SA is a variance-based simulated annealing (Var_SA) method, which uses a simulated annealing algorithm to minimize the variance merit-function for extracting a cancer-miRNA-gene module. Similarly, MSR_SA is an MSR-based simulated annealing (MSR_SA) method, which uses a simulated annealing algorithm to minimize the MSR merit-function for extracting a cancer-miRNA-gene module. The compari-son results are given in S25 Table and show that TSCCA is superior to the other tri-clustering methods in terms of multiple biological indicators and modularity score. Due to the definition of MSR, the MSR_SA method is very consuming time. We found that MSR_SA took an hour to identify a module, while Var_SA only takes 5 seconds on a personal computer. Compared with the TSCCA and Modularity_SA, the sub-tensors/modules identified by Var_SA or MSR_SA tend to be zero patterns (S6 Fig). We found that Modularity_SA has good perfor-mance results in terms of the number of cancer genes and miRNAs, while TSCCA is better in terms of the modularity score and the number of gene-gene and miRNA-gene edges (S25 Table). In addition, we also compared the performance of TSCCA and Modularity_SA under the same input data. Compared with Modularity_SA, TSCCA obtained higher modularity scores and consumed less time (S7(A) and S7(B) Fig). Therefore, from the perspective of maxi-mizing the modularity score, TSCCA is still better than the Modularity_SA.

Finally, we also compared TSCCA with principal component analysis which is applied to a joint miRNA and gene expression data from 33 TCGA cancer types (S8 Fig and S26 Table). More details and results about the comparison of TSCCA with other methods are given in S1 Text.

## Discussion

Many large projects (e.g., TCGA) have complied large multi-omics data and provided an unprecedented opportunity for deep understanding of the fundamental mechanism of cancer [51–53]. To build the connections between miRNA-gene regulatory modules across different cancer types, we developed TSCCA to identify cancer-specific and shared miRNA-gene modules using the matched miRNA and gene expression data from multiple cancers.

We applied TSCCA to the matched miRNA and mRNA expression profiles across 33 cancer types with 9,645 cancer samples for detecting cancer-related miRNA-gene modules. We found that the correlations of miRNA-miRNA, gene-gene and miRNA-gene within each module are significantly higher than those of random ones. Furthermore, we also investigated the cooperation mechanisms of miRNAs and genes within each module from multiple views: 1) whether miRNAs within the module tend to be in the miRNA family; 2) whether genes within the module tend to be enriched in some known functional classes, and whether they tend to have significantly enriched interactions in the gene interaction network; 3) whether miRNAs and genes within the module tend to have significantly enriched miRNA-gene interactions in the miRNA-gene network; 4) whether genes and miRNAs within the module tend to be cancer-related makers. We eventually found that most of the modules identified by TSCCA have cooperative characteristics or cancer-related biological functions.

We also revealed that the miRNA-gene co-expressed patterns of these identified modules show some different patterns (S2 Fig). Interestingly, a large number of miRNA-gene co-expressed patterns with positive correlation coefficients were identified, which were also observed before [54]. These results show that 1) miRNA-gene correlation patterns are heterogeneous for different cancers; 2) There may be a large number of indirect miRNA-gene regulatory relationships within each module. Furthermore, our analysis implies that these miRNA-gene patterns take different forms in different cancers. They are strongly co-expressed in some cancers while being weak in others. We also found that the miRNA-gene co-expressed patterns of some modules are reversed in different cancers. For example, the miRNA-gene correlation coefficients within module 49 are almost negative in most cancer types, while they are mostly positive in TGCT and UCS (S2(J) Fig). This observation implies the complexity of miRNA-gene regulation in cancer. Interestingly, we also found that some miRNA-gene modules can be used as diagnostic makers in different cancers. Some cancers share common survival-related modules while the others are specific to certain modules. Additionally, some cancer-specific or shared survival-related miRNAs were also found (S18 Table). This finding suggests that it is possible to develop miRNA-targeted drugs to treat multiple cancers.

In this study, we have addressed a number of important challenges in the integrative analysis of multi-omics data across multiple cancers. Some further studies are deserved to investigate in the future. First, how to extend our linear model to identify non-linear relationships between miRNAs and genes across cancer types. Second, how to integrate prior information on the relationships between genes or miRNAs (e.g., the PPI network and gene pathway) to identify more biologically meaningful patterns. Third, how to make use of other omics data, such as copy number variation and DNA methylation data. The last but not the least, how to apply our approach to other biological problems. For example, GDSC and CCLE have released

a wealth of drug and gene expression data across different cell lines [55–57]. This provides new opportunities to discover cell-specific and shared gene-drug co-modules using TSCCA.

## Supporting information

**S1 Text. Supporting methods and results.**
(DOCX)

**S1 Fig. Convergence analysis of 50 modules identified by TSCCA on the TCGA dataset across 33 cancer types.**
(PDF)

**S2 Fig. Heatmap of cancer-miRNA-gene modules identified by TSCCA in the TCGA dataset.** Each subfigure corresponds to an identified module and a random module. In each subfigure, the top half corresponds to the identified module (row corresponds to gene, column corresponds to miRNA) and the lower part of is a random module for comparison. (A) Showing the Heatmap of modules 1 to 5. (B) Showing the Heatmap of modules 6 to 10. (C) Showing the Heatmap of modules 11 to 15. (D) Showing the Heatmap of modules 16 to 20. (E) Showing the Heatmap of modules 21 to 25. (F) Showing the Heatmap of modules 26 to 30. (G) Showing the Heatmap of modules 31 to 35. (H) Showing the Heatmap of modules 36 to 40. (I) Showing the Heatmap of modules 41 to 45. (J) Showing the Heatmap of modules 46 to 50.
(PDF)

**S3 Fig. Characteristics of modules in different cancers.** (A) Heatmap showing the output matrix $W$ of Algorithm 2, when it was applied to the TCGA data. Each column corresponds to a module and each row corresponds to a cancer type and $W_{ij}$ reflects the co-expressed intensity between the genes and the miRNAs within the module $j$ on the cancer $i$. A hierarchical clustering method was used to cluster the rows (cancer types) into four clusters. (B) Scatter plot for elements of the $W$ matrix. There are three negative elements/pairs in $W$, where (Module 31, TGCT) is −0.145, (Module 49,TGCT) is −0.23 and (Module 49, UCS) is −0.138 and (C) Their heatmaps shown in the blue frame.
(PDF)

**S4 Fig. Application of the TSCCA onto the subset of TCGA cancer data from the cluster 3 in Fig 4 and extract 50 modules.** We first extracted a subset of cancers (A) and then re-used TSCCA to extract 50 modules on the subset of the previous data, and we found some new modules with significant modularity scores (B). Finally, we show the heatmap of the corresponding $W$ matrix (C).
(PDF)

**S5 Fig. miRNA-gene regulatory network analysis of modules.** (A) For each identified module, a produce is developed to identify a largest connected subgraph, i.e., a three-layer miRNA-gene regulatory network, where the miRNA-gene interactions are from miRTarBase network and the gene-gene interactions are from the gene interaction network, and miRNAs regulate genes and these genes regulate the other genes with three-layer network. (B) A miRNA-gene network contains 3619 experimentally verified miRNA-gene interactions from miRTarBase network via combing all genes and miRNAs of modules identified by TSCCA (Hypergeometric test $P = 3.5e\text{-}43$).
(PDF)

**S6 Fig. Heatmap of cancer-miRNA-gene modules identified by different methods in the TCGA dataset.** The top half of each heatmap corresponds to the module 1 (row corresponds

to gene, column corresponds to miRNA) and the lower part is a random module for comparison.
(PDF)

**S7 Fig. Comparison of different methods on the TCGA data in terms of Modularity score (A) and time (B).** We also compared the running time of different methods on a personal laptop. Box-plots show results in terms of modularity scores and running time of algorithm based on 50 different initializations of each method.
(PDF)

**S8 Fig. Results of pcModule.** (A) Heatmap of pcModule. The top half of each heatmap corresponds to the module 1 (row corresponds to gene, column corresponds to miRNA) and the lower part is a random module for comparison. (B) Comparison of modularity scores of pcModule and TSCCA modules.
(PDF)

**S9 Fig. Heatmap of some modules identified by TSCCA in the TCGA dataset.** (A) Heatmap of modules 1, 4 and 10. (B) Heatmap of modules 5, 8 and 9. (C) Heatmap of cancer-miRNA-gene module 31 identified by TSCCA in the TCGA dataset. Module 31 is a TGCT-cancer-specific miRNA-gene co-expressed module.
(PDF)

**S1 Table. The list of 7889 significant different expression genes with BH adjusted $P < 0.05$ in at least 15 cancer types.**
(XLSX)

**S2 Table. Summary of the TCGA data.**
(XLSX)

**S3 Table. Objective function values (Singular values) of modules identified by TSCCA.**
(XLSX)

**S4 Table. Cancer types and weights of modules identified by TSCCA.**
(XLSX)

**S5 Table. miRNA members and weights of modules identified by TSCCA.**
(XLSX)

**S6 Table. Gene members and weights of modules identified by TSCCA.**
(XLSX)

**S7 Table. Summary of modules concerning gene names, miRNA names and cancer type names.**
(XLSX)

**S8 Table. Significant overlap between two miRNA-gene-cancer modules/subtensors in a binary form.**
(XLSX)

**S9 Table. Modularity values for different cancer types.**
(XLSX)

**S10 Table. Enrichment analysis of modules in terms of cancer miRNAs, cancer genes, PPIs and miRNA-gene interactions.**
(XLSX)

**S11 Table. Number of significant terms.**
(XLSX)

**S12 Table. Significant GOBP terms.**
(XLSX)

**S13 Table. Significant KEGG terms.**
(XLSX)

**S14 Table. Significant Reactome terms.**
(XLSX)

**S15 Table. Module miRNAs are cooperative within miRNA families.**
(XLSX)

**S16 Table. Largest connected subnetwork (LCS) of modules where each edge is from verified miRNA-gene and gene-gene interactions.**
(XLSX)

**S17 Table. Prognostic miRNA-gene module biomarkers in multiple cancer types.**
(XLSX)

**S18 Table. Prognostic miRNA biomarkers in multiple cancer types.**
(XLSX)

**S19 Table. Biological functional analysis of selected cancer-miRNA-gene modules.**
(XLSX)

**S20 Table. Comparison (in terms of CE ± std) on the simulated data.**
(XLSX)

**S21 Table. Comparison (in terms of Recovery ± std) on the simulated data.**
(XLSX)

**S22 Table. Comparison (in terms of CE ± std) on the simulated data with different variances.**
(XLSX)

**S23 Table. Comparison of the (in terms of Recovery ± std) on the simulated data with different variables.**
(XLSX)

**S24 Table. Performance comparison of TSCCA and SCCA, where we applied SCCA to identify 50 modules on each cancer data set.**
(XLSX)

**S25 Table. Performance comparison of TSCCA and the triclustering methods.**
(XLSX)

**S26 Table. Results of pcModule.**
(XLSX)

**S27 Table. Gene-gene interaction set enrichment for the identified modules by TSCCA on the TCGA data.**
(XLSX)

**S28 Table. miRNA-gene interaction set enrichment for the identified modules by TSCCA on the TCGA data.**
(XLSX)

## Author Contributions

**Conceptualization:** Wenwen Min, Tsung-Hui Chang, Shihua Zhang, Xiang Wan.

**Data curation:** Wenwen Min, Shihua Zhang.

**Formal analysis:** Wenwen Min, Shihua Zhang, Xiang Wan.

**Funding acquisition:** Wenwen Min, Shihua Zhang, Xiang Wan.

**Investigation:** Wenwen Min, Tsung-Hui Chang, Shihua Zhang, Xiang Wan.

**Methodology:** Wenwen Min, Tsung-Hui Chang, Shihua Zhang, Xiang Wan.

**Project administration:** Wenwen Min, Shihua Zhang, Xiang Wan.

**Resources:** Wenwen Min, Shihua Zhang, Xiang Wan.

**Software:** Wenwen Min, Tsung-Hui Chang.

**Supervision:** Tsung-Hui Chang, Shihua Zhang, Xiang Wan.

**Validation:** Wenwen Min, Shihua Zhang, Xiang Wan.

**Visualization:** Wenwen Min, Shihua Zhang.

**Writing – original draft:** Wenwen Min, Tsung-Hui Chang, Shihua Zhang, Xiang Wan.

**Writing – review & editing:** Wenwen Min, Tsung-Hui Chang, Shihua Zhang, Xiang Wan.

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
