## [Decision Letter · Decision Letter 0]

2 Oct 2020

Dear Dr. Min,

Thank you very much for submitting your manuscript "TSCCA: A tensor sparse CCA method for detecting microRNA-gene patterns from multiple cancers" for consideration at PLOS Computational Biology.

As with all papers reviewed by the journal, your manuscript was reviewed by members of the editorial board and by several independent reviewers. In light of the reviews (below this email), we would like to invite the resubmission of a significantly-revised version that takes into account the reviewers' comments.

Please apologise the slow turnaround. The paper has now been assessed by two experts in the field who found your work of interest but have flagged a number of major issues. If you feel you can address these criticisms, concerning a comparison to other triclustering methods and software implementation, but also in relation to your simulation study, then we would invite you to resubmit an appropriately revised manuscript.

We cannot make any decision about publication until we have seen the revised manuscript and your response to the reviewers' comments. Your revised manuscript is also likely to be sent to reviewers for further evaluation.

Sincerely,

Moritz Gerstung

Guest Editor

PLOS Computational Biology

Florian Markowetz

Deputy Editor

PLOS Computational Biology

Please apologise the slow turnaround. The paper has now been assessed by two experts in the field who found your work of interest but have flagged a number of major issues. If you feel you can address these criticisms, we would invite you to resubmit an appropriately revised manuscript.

Reviewer's Responses to Questions

**Comments to the Authors:**

Reviewer #1: The paper presents TSCCA, a novel algorithm for identification of miRNA-gene modules across multiple cancer types from TCGA. TSCCA is based on CCA and extends it to tensors to support several cancer types. The authors apply the method to 33 cancers from TCGA, and show the biological relevance of the detected modules using several databases of biological knowledge, and by considering the survival implications of the selected genes and miRNA. The authors also compare TSCCA to Modularity SA and to SCCA on cancer and synthetic data.

The task of detecting miRNA-gene modules in cancer is of high importance, given the known involvement of miRNA in cancer and the complex regulatory processes that are perturbed in cancer. We are not familiar with previous methods designed to find miRNA-gene modules across several datasets, so this method has potentially high value. The paper is well organized and well written. However, as previous methods detected miRNA-gene modules, and the main innovation of this method is in working on multiple datasets, the benefit of using multiple cancers should be shown in a more convincing manner. Additionally, while no algorithm was previously developed for the specific data used in the study, the problem of tri-clustering (detecting modules in tensors) was previously investigated, and the suggested method should be compared to other tri-clustering methods, in addition to modularity SA. Furthermore, the authors should provide an implementation of TSCCA. If these points are satisfactorily addressed, we think that this work would merit publication in PLOS Computational Biology.

Major comments

1. Previous methods were developed to detect miRNA-gene modules, and the main novelty of TSCCA is that it uses multiple datasets. The advantage of using multiple datasets is not clearly demonstrated. The criteria used to show the biological and clinical relevance of the detected modules (enrichment of cancer genes and miRNAs in the modules, enrichment of miRNA families, miRNA-gene regulatory networks, and survival analysis using the modules) are never compared to solutions obtained on a single cancer dataset. It is very likely that all of these criteria would also show biological relevance if SCCA would be applied to each cancer separately.

The multi-dataset nature of the algorithm is currently considered in two places. The first is Figure 4, where the W matrix is visualized. This matrix shows that the modules actually capture only a subset of the cancer types, while other cancer types have many zero or near-zero loadings in W. To these reviewers, it seems surprising that all these cancer types have such a small number of miRNA-gene modules. Is this a biological reality or a bias introduced by the algorithm? The authors should rerun TSCCA after excluding the few cancer types that are responsible for most of the modules, and examine the output of the algorithm. If new biologically significant modules emerge, this would suggest that dominant caner types overshadow the others. In this case this limitation of the algorithm should be clearly stated.

The second place where the multi-dataset aspect is considered is the direct comparison to SCCA. This analysis is more convincing, but it should be expanded in two ways. First, biological criteria should be used for the comparison (enrichment of cancer genes etc.), and not only the modularity. Second, it is interesting to show the modularity when calculated only on one cancer type (e.g. ACC) when TSCCA is applied to all the data, and SCCA is applied only on ACC. This can show when using one dataset to detect miRNA-gene modules is preferable to using multiple datasets, highlighting the limitations of the algorithm.

2. To the best of our knowledge, no previous method was developed to detect miRNA-gene modules across multiple datasets, but other methods were developed for a similar computational task – triclustering. The only triclustering algorithm TSCCA is compared to is Modularity_SA, which is a method that the authors developed themselves for the comparison. The authors should compare TSCCA to other triclustering methods, even if they do not perform l0 regularization. A survey of triclustering algorithms can be found in [1].

As stated in the previous point, the comparison should include biological criteria. The authors demonstrate convincingly that TSCCA's modules are biologically relevant, but biological criteria (e.g. enrichment of gene-gene interactions) should be compared to other methods.

3. The authors should provide an implementation (or at least a well documented executable) of TSCCA, as the method, rather than the biological results, are the main contribution of this work.

Other important comments

4. In order to optimize TSCCA's objective function (equation (5)), the authors optimize for u, v and w separately. If w is held constant, the optimization problem is that of sparse diagonal CCA, which is solved by the work of Xu et al. that the authors cite (reference [38] in the paper). Their algorithm can be used for an improved optimization procedure.

5. The analysis for whether the 50 modules significantly overlap is not clear.

a. Does the number of overlapping elements include genes, miRNA and cancer types? If so, it places a higher emphasis on gene expression, which is the most common feature. Or are three tests for overlap performed separately, one for genes, one for miRNA and one for cancer types? If the latter, how are these tests integrated?

b. The random modules are created by sampling 100 genes, 10 miRNA and 20 cancers. While these are the parameters used to run TSCCA, the authors state that not all modules include this number of genes, miRNA and cancers. The number of features in these random modules is therefore different from the number of features in the original modules, and the distribution of the size of the overlap is also different. The tests should be performed by conditioning on the size of the modules.

c. There seems to be a small error in the number of random modules. The appendix says 100 modules were generated, but the number of module pairs appears to be 1000 in the rest of the text.

6. In a couple of analyses the authors do not statistically show the merit of their results.

a. The authors count the number of modules with at least two miRNAs from the same family. Though 46 out of 50 sounds high, a statistical test should be performed to derive a p-value.

b. The authors found that 70% of the modules have at least 3 miRNAs participating in a three-layer network, but the significance of this observation is not clear without a statistical test.

7. The geometric tests used to calculate gene-gene and miRNA-gene interaction set enrichment (as described in sections 15 and 16 of the appendix) do not condition on the degree of the genes and miRNAs in the networks. In the way these tests are currently performed, it is possible to obtain significant p-values just because a chosen gene has high degree in the gene-gene network (or similarly, a gene / miRNA has high degree in the gene-miRNA network), even if there's no enrichment of interaction in the module. Indeed, cancer genes are generally known to have high degrees in gene-gene networks. The tests should be performed by permuting genes / miRNA conditioned on their degree in the networks.

8. The CE and Recovery score does not consider the cancers selected for each module. The authors need to add an additional metric (or change the current one) that considers the cancers. Additionally, because the number of miRNAs is smaller than the number of genes, it is of interest to report the CE and Recovery score when considering only genes or only miRNAs. Otherwise the current reporting places more emphasis on genes.

9. The work will be much improved if the biology behind several selected modules is described. How do the discovered modules improve our understanding of pan-cancer gene and miRNA regulation?

Minor comments

10. Prior work by Tan et al. recently investigated miRNA-gene modules across cancers [2]. Their computational approach is quite naïve, but this work should be mentioned.

11. Though the appendix states where corrections for multiple hypotheses were performed, we think that for clarity, these corrections should be also mentioned in the main text when they are used, and whether reported p-values are after correction.

12. The external datasets used in this study are mentioned at the beginning, but it would be helpful if they are also mentioned when they are being used. For example, in section 3.3 it would be helpful if the databases for cancer genes and miRNA are mentioned.

13. The text describing Figure 3C is not clear. It resembles the text for 3B, even though these two examples demonstrate very different phenomena.

14. The objective functions for TSCCA allows for negative W values. It is therefore interesting to visualize W, in addition to the current visualization in Figure 4 of |W|. Are there miRNA-gene modules that are correlated in one cancer type, but are anti-correlated in another? This is briefly discussed in the discussion, and is seen in the supplementary figures, but will be more easily shown by visualizing W directly.

15. "For each miRNA-gene module, … have not been verified" – this sentence is not clear. What verification was performed?

16. Because each module has more genes than miRNA, the 1st PC will likely mainly represent the variance in gene expression. It will be interesting to repeat this analysis by taking the 1ST PC using genes only, or the 1st PC using miRNA only.

17. When describing the survival analysis, modules 11 and 36 are specifically mentioned. We understand that this is because these modules had non-zero entries in W in the cancer types for which they were linked to survival. This should be explicitly stated, as otherwise it is not clear why these modules are mentioned.

18. Typos:

a. "For each cancer types, we downloaded..." – should be "type" (p. 3).

b. "We found that 7889…" – "that" should be removed (p. 3).

c. In equation (6), the l0 constraint should be on v, not on u.

d. "While we also found that the modularity…" – remove "While" (p. 7).

e. "negative correlation on all cancers on all cancers". (p. 9).

f. "from a experimentally validated…" – replace "a" with "an" (p. 12).

g. When describing the simulation, the distribution for A_3 is written twice. The second time should be A_4 (p. 15).

h. "it miss some real members" – should be "misses" (p. 15). There are several times "miss" should be replaced with "misses" in this page.

References

[1] - Rui Henriques and Sara C. Madeira. 2018. Triclustering Algorithms for Three-Dimensional Data Analysis: A Comprehensive Survey. ACM Comput. Surv. 51, 5, Article 95 (January 2019), 43 pages. DOI:https://doi.org/10.1145/3195833

[2] - Hua Tan, Shan Huang, Zhigang Zhang, Xiaohua Qian, Peiqing Sun, and Xiaobo Zhoua. 2019. Pan-cancer analysis on microRNA-associated gene activation. EBioMedicine. 2019 May; 43: 82–97. doi: 10.1016/j.ebiom.2019.03.082

Reviewer #2: To identify cancer-specific and shared miRNA-gene co-expressed modules, the authors proposed a tensor sparse canonical correlation analysis (TSCCA) method to analysis of matched miRNA and gene expression data of multiple cancers. The authors first constructed a tensor of gene, miRNA and cancers. Then they decomposed the correlation tensor into a number of latent factors. Finally, based on the non-zero latent factors, they identified cancer-miRNA-gene modules. Application to 33 TCGA cancer types identified novel cancer-related miRNA-gene modules. Here are my comments:

1.In simulation, the data was generated from normal distributions with fixed variances. I wonder what would happen if the values were generated with larger variances, instead of 0.04?

2.It is claimed that TSCCA method can identify both cancer-specific and shared miRNA-gene co-expressed modules. I wonder whether TSCCA identified any shared miRNA-gene co-expressed modules across 33 TCGA cancer types. If so, what properties do the shared modules have?

3.The authors evaluated the performance of TSCCA with other methods on the TCGA data mainly using the modularity score. They may want to evaluate the modules identified by the comparison methods using other measurements, for example, they may check the enrichment of cancer related genes/miRNAs among these modules.

**Have all data underlying the figures and results presented in the manuscript been provided?**

Reviewer #1: None

Reviewer #2: None

PLOS authors have the option to publish the peer review history of their article (what does this mean?). If published, this will include your full peer review and any attached files.

Reviewer #1: No

Reviewer #2: No
---

## [Decision Letter · Decision Letter 1]

28 Jan 2021

Dear Dr. Min,

Thank you very much for submitting your manuscript "TSCCA: A tensor sparse CCA method for detecting microRNA-gene patterns from multiple cancers" for consideration at PLOS Computational Biology.

As with all papers reviewed by the journal, your manuscript was reviewed by members of the editorial board and by several independent reviewers. In light of the reviews (below this email), we would like to invite the resubmission of a significantly-revised version that takes into account the reviewers' comments, especially those raised by reviewer 1.

We cannot make any decision about publication until we have seen the revised manuscript and your response to the reviewers' comments. Your revised manuscript is also likely to be sent to reviewers for further evaluation.

Sincerely,

Moritz Gerstung

Guest Editor

PLOS Computational Biology

Florian Markowetz

Deputy Editor

PLOS Computational Biology

Thank you for submitting a revised manuscript. As you will see, reviewer 1 continues to raise a series of major concerns that need to be addressed to make the manuscript suitable for publication.

Reviewer's Responses to Questions

**Comments to the Authors:**

Reviewer #1: The authors made a thorough revision and seriously considered our previous remarks. This significantly improved the paper. The three main issues we pointed to previously have been largely addressed. We do have several more comments, but if they are addressed as well, the paper is worthy of publications.

Major comment

The one-sample Wilcoxon signed-rank test is used several times in the manuscript, e.g. in the analysis in section 3.2. We do not think this test is appropriate here. The null hypothesis for the Wilcoxon test is that the modularity score for a module is higher than the median for the randomly generated modules. This p-value can be very low even if the module’s modularity score is not very extreme in comparison to the modularity scores of the random modules. For example, consider a module with modularity score 2000, and 1000 random modules with scores 1401, 1402, …, 2400. The p-value for Wilcoxon will be very small (<2e-16 according to our test in R), because we can confidently say that the modularity of the original module (2000) is higher than the median modularity of the random modules. However, 40% of the random modules have a higher score. A better way to calculate an empirical p-value is to count the number of randomly permuted modules with modularity score equal or greater than 2000.

In the same section (3.2), it is not clear how a single p-value is calculated. Section S8 in the appendix only describes how to calculate a p-value for a single module, and it is not clear how these p-values are integrated into a single one (which is stated in the paper as “P < 0.001”).

This comment doesn’t apply only to section 3.2, but to other cases in which the one-sample Wilcoxon signed-rank test is used (e.g. the number of modules with at least 2 miRNAs from the same family).

Other important comments

- To perform permutation testing while conditioning on the degrees in a network, the authors sample genes such that the sum of their degrees is close to the sum of degrees in the original gene set. This conditions on the sum of degrees, but not on the full degree distribution. The common way to perform permutations while conditioning on the degree is to permute gene names only between genes with the same degree. For example, a gene with degree 5 can only be replaced in the permutation with another gene of degree 5. (If sample is insufficient, this can be done by forming bins of genes by the degree, and permuting between genes from the same bin)

- Two of our major comments were that further comparison to triclustering methods is required, and that the advantage of using multiple cancers is not sufficiently shown. The authors addressed these points, but some of the results from these new analyses are only mentioned briefly in the discussion, while they should be stated more clearly:

o The W matrix visualization shows that a few cancers dominate all the created modules. In the discussion the authors mention this, and show another analysis in which these dominant cancers are removed (in appendix figure S16). This point, that a small number of cancers may dominate the results, is a major caveat of the analysis, and as such it should be mentioned when first presenting TSCCA’s results, and not briefly referred to in the discussion.

o Modularity_SA has very good results in terms of the number of cancer genes and miRs, while TSCCA is better in terms of the number of gene-gene and miR-gene edges. The good performance of Modularity_SA, and its advantages in comparison to TSCCA, should be mentioned in the main text.

Minor comment

In tables reporting the results of TSCCA, the font in the rows representing TSCCA’s results is bold. It is common practice to mark in bold the best result in each column, and we suggest the authors do the same here, or remove the bold font from TSCCA’s row. Otherwise it looks as if TSCCA always has the best performance.

Typos:

- “within this module were verified reported before” (p. 12) – remove “verified” or “reported”.

- “We assessed the similarity of between the true modules” (p. 13) – remove “of”.

- “miRNA-gene correlation patterns are heterogeneity” (p. 15) – should be “heterogeneous”.

- “explorative tool, which identity” (p. 15) – should be “identify”.

Reviewer #2: The authors conducted simulation and real data studies to address the questions. There is a question about Table S25. The authors compared TSCCA with other methods using multiple biological indicators and modularity score and concluded TSCCA is superior to the other tri-clustering methods. However, Modularity_SA identified more cancer_miR and cancer_gene than the TSCCA. The authors may want to discuss this before directly concluding that TSCCA is superior to the other tri-clustering methods.

**Have all data underlying the figures and results presented in the manuscript been provided?**

Reviewer #1: Yes

Reviewer #2: None

PLOS authors have the option to publish the peer review history of their article (what does this mean?). If published, this will include your full peer review and any attached files.

Reviewer #1: No

Reviewer #2: No
---

## [Decision Letter · Decision Letter 2]

5 May 2021

Dear Dr. Min,

We are pleased to inform you that your manuscript 'TSCCA: A tensor sparse CCA method for detecting microRNA-gene patterns from multiple cancers' has been provisionally accepted for publication in PLOS Computational Biology.

Best regards,

Moritz Gerstung

Guest Editor

PLOS Computational Biology

Florian Markowetz

Deputy Editor

PLOS Computational Biology

Reviewer's Responses to Questions

**Comments to the Authors:**

Reviewer #1: PCOMPBIOL-D-20-01172: TSCCA: A tensor sparse CCA method for detecting microRNA-gene patterns from multiple cancers

Wenwen Min, Ph.D.; Tsung-Hui Chang; Shihua Zhang; Xiang Wan

The authors considered all our previous remarks, and we now consider this paper as worthy of publication.

We only have a couple of minor suggestions:

- Even though the analysis the authors performed in section 3.2 is now clear, its phrasing is not (this refers to the sentences "The identified modules with… random ones (permutation test P < 0.05 / 50)"). We understand that all modules were statistically significant, but this is not directly stated (it currently says "these modules" rather than "all modules"). Also, k should be removed, and the p-value of 0.05 / 50 mentioned only once. Since it is currently mentioned twice, it seems as if it relates to two different analyses.

- Section 3.4: "These modularity scores… those of the random ones". This sentence is not backed by any statistical analyses, but only by two examples in Figure 5A and 5E. Either a statistical analysis should be performed to validate this sentence, or it should be explicitly stated that this sentence only refers to two examples.

Reviewer #2: No more comments.

**Have the authors made all data and (if applicable) computational code underlying the findings in their manuscript fully available?**

Reviewer #1: Yes

Reviewer #2: Yes

PLOS authors have the option to publish the peer review history of their article (what does this mean?). If published, this will include your full peer review and any attached files.

Reviewer #1: No

Reviewer #2: No

---

## [Editor Report · Acceptance letter]

24 May 2021

PCOMPBIOL-D-20-01172R2 

TSCCA: A tensor sparse CCA method for detecting microRNA-gene patterns from multiple cancers

Dear Dr Min,

I am pleased to inform you that your manuscript has been formally accepted for publication in PLOS Computational Biology. Your manuscript is now with our production department and you will be notified of the publication date in due course.

With kind regards,

Katalin Szabo
